# Systemic and Lower Respiratory Tract Immunity to SARS-CoV-2 Omicron and Variants in Pediatric Severe COVID-19 and Mis-C

**DOI:** 10.3390/vaccines10020270

**Published:** 2022-02-10

**Authors:** Juanjie Tang, Adrienne G. Randolph, Tanya Novak, Tracie C. Walker, Laura L. Loftis, Matt S. Zinter, Katherine Irby, Surender Khurana

**Affiliations:** 1Center for Biologics Evaluation and Research (CBER), Division of Viral Products, Food and Drug Administration (FDA), Silver Spring, MD 20993, USA; jessicatang217@gmail.com; 2Department of Anesthesiology, Critical Care and Pain Medicine, Department of Anesthesia, Harvard Medical School, Boston Children’s Hospital, Boston, MA 02115, USA; adrienne.randolph@childrens.harvard.edu (A.G.R.); tanya.novak@childrens.harvard.edu (T.N.); 3Department of Pediatrics, University of North Carolina at Chapel Hill Children’s Hospital, Chapel Hill, NC 27514, USA; twalker3@med.unc.edu; 4Department of Pediatrics, Division of Critical Care Medicine, Baylor College of Medicine, Houston, TX 77030, USA; llloftis@texaschildrens.org; 5Department of Pediatrics, Divisions of Critical Care and Bone Marrow Transplantation, University of California, San Francisco, CA 94158, USA; matt.zinter@ucsf.edu; 6Department of Pediatrics, Division of Pediatric Critical Care Medicine, Arkansas Children’s Hospital, Little Rock, AR 72202, USA; okvaden@uams.edu

**Keywords:** SARS-CoV-2, COVID-19, pediatric, MIS-C, mucosal immunity, neutralization, trachea, omicron, variants

## Abstract

Mucosal immunity plays an important role in the control of viral respiratory infections like SARS-CoV-2. While systemic immune responses against the SARS-2-CoV-2 have been studied in children, there is no information on mucosal antibody response, especially in the lower respiratory tract of children coronavirus disease 2019 (COVID-19) and post-infectious multisystem inflammatory syndrome in children (MIS-C) against emerging SARS-CoV-2 variants. Therefore, we evaluated neutralizing antibody responses in paired plasma and endotracheal aspirates of pediatric severe, acute COVID-19 or MIS-C patients against SARS-CoV-2 WA1/2020, as well as against variants of concern (VOCs). Neutralizing antibody responses against the SARS-CoV-2 WA1/2020 strain in pediatric plasma were 2-fold or 35-fold higher compared with the matched endotracheal aspirate in COVID-19 or MIS-C patients, respectively. In contrast to plasma, neutralizing antibody responses against the VOCs and variants of interest (VOIs) in endotracheal aspirates were lower, with only one endotracheal aspirate demonstrating neutralizing titers against the Iota, Kappa, Beta, Gamma, and Omicron variants. In conclusion, our findings suggest that children and adolescents with severe COVID-19 or MIS-C have weak mucosal neutralizing antibodies in the trachea against circulating SARS-CoV-2 Omicron and other VOCs, which may have implications for recovery and for re-infection with emerging SARS-CoV-2 variants.

## 1. Introduction

SARS-CoV-2 infection in children is often asymptomatic; however, children and adolescents can develop severe COVID-19 and its associated post-infectious multisystem inflammatory syndrome in children (MIS-C) [1]. Systemic antibody response against SARS-CoV-2 have been evaluated in pediatric COVID-19 and MIS-C [2,3]. However, mucosal antibody response is unknown, especially in the lower respiratory tract of children that can play critical role in protection against disease. The emergence of Omicron variant of SARS-CoV-2, and its rapid spread across the globe, resulted in designation of Omicron as a VOC. Its suggested that Omicron possibly evolved from ancestral SARS-CoV-2 around mid-2020 or the variant may have evolved in a single immunocompromised individual with long-term SARS-CoV-2 infection [4]. Omicron, contains large number of mutations including at least 30 in the SARS-CoV-2 spike protein, resulting in resistance to neutralizing antibodies generated following SARS-CoV-2 vaccination and infection [5,6,7,8]. It is important to understand the mucosal immunity in the lower respiratory tract, which is the site of virus replication resulting in severe disease, and its ability to neutralize emerging SARS-CoV-2 variants, especially in children.

Therefore, in this study we evaluated neutralizing antibody response in the lower respiratory tract of hospitalized children with severe acute COVID-19 and MIS-C and compared it with systemic immune response against the SARS-CoV-2 WA1/2020 strain, five variants of concern (VOCs) [Alpha variant (B.1.1.7), Gamma variant (P.1), Beta variant (B.1.351), Delta variant (B.1.617.2), and Omicron variant (B.1.1.529)], and three variants of interest (VOIs) [Epsilon (B.1.429), Iota (B.1.526), and Kappa (B.1.617.1)].

## 2. Methods

### 2.1. Pediatric Samples

The Overcoming COVID-19 Network studies severe complications of COVID-19 in children and adolescents under a single IRB at Boston Children’s Hospital under Protocol Number #IRB-P00033157, and informed consent was obtained from at least one parent or legal guardian. 

All pediatric patients were ≤19 years old with confirmed SARS-CoV-2 positive PCR and/or positive antibody tests. Endotracheal aspirates and paired plasma were only obtained from severe hospitalized intubated patients. Endotracheal aspirates from children with ‘mild’ cases were not available as controls, since majority of pediatric patients with MIS-C or acute COVID-19 do not need intubation. Endotracheal aspirates were suctioned from mechanically ventilated patients using a 6 Fr. catheter attached to the standard trap, kept on ice during transfer to the lab where it was vortexed and then centrifuged at 2500× *g* for 10 min. Supernatant was removed and frozen at −80 °C. 

Patients with immune compromising conditions that could impair antibody responses were excluded, as were patients with life support limitations or end stage lung disease. Hospitalized patients with COVID-19-related complications were hospitalized for acute COVID-19 or multisystem inflammatory syndrome in children (MIS-C) and samples were collected within a median of 64 hrs of hospital admission. MIS-C and acute COVID-19 were defined using U.S. Centers for Disease Control and Prevention (CDC) case definitions. Acute COVID-19 was defined as having signs or symptoms that could be associated with early SARS-CoV-2 infection accompanied by a positive real-time polymerase chain reaction (RT-PCR) test for severe acute respiratory syndrome coronavirus 2 (SARS-CoV-2) as defined by the U.S. Centers for Disease Control and Prevention (CDC) as listed on their website (https://wwwn.cdc.gov/nndss/conditions/coronavirus-disease-2019-covid-19/case-definition/2020/ accessed on 5 April 2020). MIS-C patients met the criteria for Multisystem Inflammatory Syndrome in Children as defined by the CDC as listed on their website (https://www.cdc.gov/mis-c/hcp/ published on 14 May 2021).

Samples were tested in different antibody assays with approval from the U.S. Food and Drug Administration’s Research Involving Human Subjects Committee (FDA-RIHSC) under exemption protocol -252-Determination- CBER-19 August 2020. All samples were heat treated at 56 °C for 1 h.

### 2.2. Quantification of Total IgM, IgG, and IgA in Pediatric Samples

The amount of total IgM, IgG, and IgA in plasma and endotracheal aspirates was quantified using human immunoglobulin quantitation kit (ThermoFisher, Waltham, MA, USA) as per the manufacturer’s instructions.

### 2.3. SARS-CoV-2 Neutralization Assay

Plasma and ET samples were evaluated in a qualified SARS-CoV-2 pseudovirion neutralization assay (PsVNA) using the SARS-CoV-2 WA1/2020 strain, 5 variants of concern (VOCs) [Alpha variant (B.1.1.7), Gamma variant (P.1), Beta variant (B.1.351), Delta variant (B.1.617.2), and Omicron variant (B.1.1.529)], and 3 VOIs [Epsilon (B.1.429), Iota (B.1.526), and Kappa (B.1.617.1)] (Appendix A). SARS-CoV-2 neutralizing activity measured by PsVNA correlated with PRNT (plaque reduction neutralization test with authentic SARS-CoV-2 virus) in previous studies [9,10,11].

Pseudovirions were produced as previously described [11]. Briefly, human codon-optimized cDNA encoding the SARS-CoV-2 S glycoprotein of the WA1/2020 and the variant strain were synthesized by GenScript and cloned into eukaryotic cell expression vector pcDNA 3.1 between the *BamH*I and *Xho*I sites. Pseudovirions were produced by co-transfection Lenti-X 293T cells with psPAX2 (gag/pol), pTrip-luc lentiviral vector, and pcDNA 3.1 SARS-CoV-2-spike-deltaC19, using Lipofectamine 3000. The supernatants were harvested at 48 h post transfection and filtered through 0.45 µm membranes and titrated using 293T-ACE2-TMPRSS2 cells (HEK 293T cells that express ACE2 and TMPRSS2 proteins) [11].

Neutralization assays were performed as previously described [3,9,12]. For the neutralization assay, 50 µL of SARS-CoV-2 S pseudovirions (counting ~200,000 relative light units) were pre-incubated with an equal volume of medium containing plasma at serial dilutions at room temperature for 1 h, then virus-antibody mixtures were added to 293T-ACE2-TMPRSS2 cells [11] in a 96-well plate. The input virus with all eight SARS-CoV-2 strains used in the current study were the same (2 × 10^5^ Relative light units/50 µL/well). After a 3 h incubation, fresh medium was added to the wells. Cells were lysed 24 h later, and luciferase activity was measured using One-Glo luciferase assay system (Promega, Madison, WI, USA, Cat# E6130). Controls included cells only, the virus without any antibody, and positive sera. The limit of detection for the neutralization assay was 1:20. Two independent biological replicate experiments were performed for each sample and variation in PsVNA50 titers was <9% between replicates.

### 2.4. Quantification and Statistical Analysis

Descriptive statistics were performed to determine the geometric mean titer values and were calculated using GraphPad. All experimental data to compare differences among groups were analyzed using lme4 and emmeans packages in R (RStudio version 1.1.463).

The demographic characteristics of these study participants are shown in Appendix A. Since age can be a biologically plausible confounder, data were analyzed for statistical significance between SARS-CoV-2 strains to control for age as covariate (predictor variables) using a multivariate linear regression model in R. To ensure robustness of the results, absolute neutralization titers were log2-transformed before performing the analysis. For comparisons between the SARS-CoV-2 strains (factor variable), pairwise comparisons were extracted using ‘emmeans’ and Tukey-adjusted p values were used for denoting significance to reduce Type 1 error due to multiple testing in R (RStudio version 1.1.463). The tests were 2-sided tests. The differences were considered statistically significant with a 95% confidence interval when the *p* value was less than 0.05. (* ≤ 0.05, ** ≤ 0.01, *** ≤ 0.001, **** ≤ 0.0001).

Correlation and regression analyses were performed by computing Spearman’s rank correlation coefficient and significance in GraphPad Prism.

## 3. Results

As bronchoalveolar lavage is almost never performed in pediatric patients with COVID-19 or MIS-C, we analyzed mucosal immunity in the trachea, since the trachea is considered as the lower respiratory tract in children [13]. Paired plasma and endotracheal aspirate (ET) samples were collected in intensive care unit from 5 acute severe COVID-19 pediatric patients (SARS-CoV-2 PCR-positive) and 5 MIS-C patients (4 PCR-positive, 4 SARS-CoV-2 antibody positive) following intubation for acute respiratory failure between June to early November 2020, prior to emergence of VOCs/VOIs in US (Appendix A). Patients were aged <1 to 19 years (median age: 10 years). The median duration of hospitalization was 13.5 days and all patients survived to discharge. Median time to ET sample collection from hospital admission was 64 h (interquartile range (IQR) 33, 97 h). None of the 10 hospitalized children were on chronic immunosuppressive drugs or chemotherapy and only one of them received IVIG without COVID-19 antibodies [9] prior to collection of paired ET/plasma samples. Neutralizing antibody responses in matched plasma and endotracheal aspirate supernatants were measured against the SARS-CoV-2 WA1/2020 strain, as well as against VOIs: Epsilon (B.1.429), Iota (B.1.526), and Kappa (B.1.617.1), and VOCs: Alpha variant (B.1.1.7), Beta variant (B.1.351), Gamma variant (P.1), Delta variant (B.1.617.2), and Omicron (B.1.1.529) using a pseudovirion neutralization assay (PsVNA) as described [9] to measure PsVNA50 (50% reduction in viral titers) (Appendix A). SARS-CoV-2 neutralizing activity measured by PsVNA correlated with PRNT (plaque reduction neutralization test with authentic SARS-CoV-2 virus) in previous studies [9].

The total IgM, IgG, and IgA levels in plasma were 40-(IgM) to 54-fold (IgG and IgA) higher than in the matched trachea of pediatric COVID-19 (Table 1). In MIS-C, the antibody levels in plasma were 44-(IgM), 75-(IgA) to 169-fold (IgG) higher than antibody levels in the trachea. All plasma samples collected either from severe COVID-19 or MIS-C contained neutralizing antibodies against the ancestral WA1/2020 strain (Figure 1A). Plasma geometric mean titers (GMT) against SARS-CoV-2 WA1/2020 in COVID-19 (1:779) were similar to those of MIS-C (1:932) patients (Appendix A). GMTs of plasma against Alpha, Epsilon, Iota, Kappa, and Delta were reduced by 4.1 to 11.4-fold, were 19.8-fold lower against Gamma (1:43), and 30.4-fold lower against Beta (1:28) and were the lowest (50.1-fold) against Omicron (1:17) compared with WA1/2020 (1:852) (Figure 1A). Plasma from MIS-C (who were exposed to SARS-CoV-2 weeks prior to hospitalization) showed lower GMT against all SARS-CoV-2 variants compared with COVID-19 patients (Appendix A).

In contrast to plasma, neutralizing antibody titers were observed in endotracheal aspirates of only 2/5 COVID-19 and 3/5 MIS-C patients against WA1/2020 strain with PsVNA50 titers ranging from 1:10 and 1:2495 (Figure 1B, Appendix A). The teen with acute COVID-19 with the highest ET neutralization titer (1:2495) against WA1/2020 was obese but otherwise healthy and was intubated 8 days with the ET sample collected on day 4 and showed high neutralizing antibodies against all the variants tested (Appendix A). PsVNA50 titers against variants were lower in ET of COVID-19 and MIS-C patients, with only one of the COVID-19 ET demonstrating neutralizing antibodies against the Iota, Kappa, Beta, Gamma, and Omicron variants (Appendix A). Overall, the endotracheal aspirates GMTs against SARS-CoV-2 and its variant strains were 2 to 35-fold lower compared with matched plasma (Appendix A). A correlation was observed between the neutralization titers of plasma with matched endotracheal aspirates against the WA1/2020 and variants (Figure 2).

## 4. Discussion

Mucosal immunity plays an important role in control as well as pathophysiological immune-mediated disorders of respiratory viral infections like SARS-CoV-2. Our data demonstrate that most children and adolescents with severe COVID-19 or MIS-C have weak mucosal neutralizing antibodies in the trachea against circulating SARS-CoV-2 Omicron and other VOCs, and therefore may be susceptible to re-infection with emerging SARS-CoV-2 variants as has been observed in recent months for infection of children with the Omicron variant [14]. These findings are consistent with the weak mucosal response observed in the upper respiratory tract of adults with either COVID-19 or upon infection with respiratory syncytial virus [12]. The great majority of pediatric patients with MIS-C or acute COVID-19 do not need intubation, therefore this cohort reflects only the smaller percentage of children and adolescents that develop severe hypoxic respiratory failure related to SARS-CoV-2. The mechanisms of SARS-CoV-2 variant escape is under investigation that may involve loss of neutralizing antibody epitopes on evolving spike, increase in ACE2 affinity of spike or additional receptor usage or potentially progressively more positive charge on the virus spike-protein resulting in increased viral binding to the negatively charged surface of the cell [6,7,8,15,16]. The low mucosal immune response in these children can be possibly due to timing of collection of samples at early time-point following hospitalization prior to the generation of strong immune response following infection. In an earlier study, high neutralizing antibodies against SARS-CoV-2 ancestral strain were observed in bronchoalveolar lavage fluid collected from SARS-CoV-2 adult patients between days 4 and 23 after symptom onset [17].

Vaccination rates in children is low, especially in children who had prior MIS-C, and hence they are potentially at risk of another hyperinflammatory episode if re-infected with a variant strain. Therefore, children with low mucosal immunity against variants may have risk of transmission to other vulnerable populations [14]. The low respiratory tract antibodies in these children could be associated with the development of long COVID sequelae such as those observed in an adolescent [18], and therefore a better understanding of systemic and especially mucosal immunity against SARS-CoV-2 is required in children. Our study underlines the importance of continued efforts to evaluate vaccines in young children, and optimal vaccination strategies that can provide protection against circulating SARS-CoV-2 strains in this highly vulnerable pediatric population. Therefore, vaccine platforms that can generate effective durable mucosal immunity both in upper and lower respiratory tract would be helpful to provide both protection from disease as well as reduce viral titers in respiratory tract and limit viral transmission to curtail the spread of SARS-CoV-2.

## 5. Conclusions

Our study for the first time describes the lower respiratory antibody response in pediatric COVID-19 and MIS-C patients against SARS-CoV-2 variants. Low mucosal immunity against Omicron and other variants suggest that these children can be re-infected with evolving variants. Therefore, it’s prudent to develop vaccines that elicit strong long-lasting mucosal immunity to prevent infection and stop transmission of new SARS-CoV-2 variants.

## Figures and Tables

**Figure 1 vaccines-10-00270-f001:**
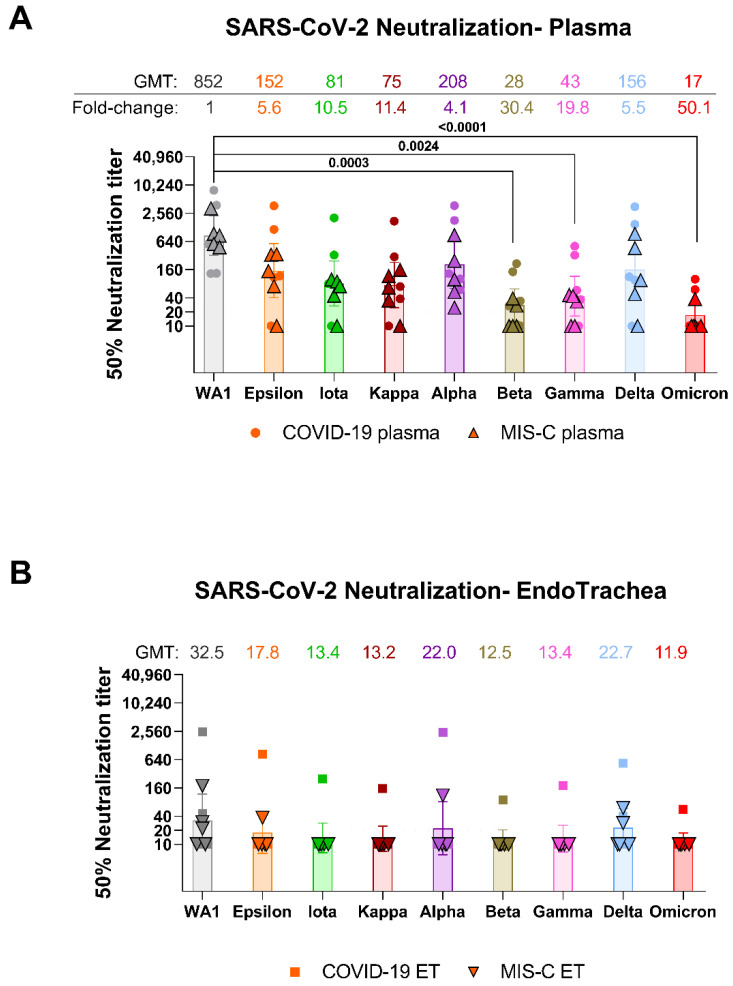
Neutralizing antibody responses in plasma and endotracheal aspirates of pediatric COVID-19 and MIS-C against SARS-CoV-2 WA1/2020 strain and variant strains. SARS-CoV-2 neutralizing antibody titers of plasma (**A**) and endotracheal aspirates (**B**) from 5 COVID-19 and 5 MIS-C pediatric patients against SARS-CoV-2 WA1/2020 strain, VOIs; Epsilon, Iota, Kappa, and VOCs; Alpha, Beta, Gamma, Delta and Omicron variant by pseudovirion neutralization assay (PsVNA). The assay of each sample was performed in duplicate to determine the 50% neutralization titer (PsVNA50). Each data point represents an individual sample (circles) and indicates the 50% neutralization titer obtained with each sample against the indicated pseudovirus. Individual COVID-19 plasma samples are shown as circles, while MIS-C plasma samples are shown as triangles in panel A. In panel B, the endotracheal aspirates samples from pediatric COVID-19 are shown as squares, while MIS-C are shown as inverted triangles. The heights of the bars and the numbers over the bars indicate the geometric mean titers, and the whiskers indicate 95% confidence intervals. The horizontal dashed line indicates the limit of detection for the neutralization assay (PsVNA50 of 20). The fold-change indicates the average decrease in neutralization titer of the indicated variants as compared with that of the SARS-CoV-2 WA1/2020 virus. The raw data and information regarding the samples for each pediatric participant (age and 50% neutralization titers against various SARS-CoV-2 strains) are summarized in Appendix A. Differences in neutralization titers between SARS-CoV-2 strains were analyzed by lme4 and emmeans packages in R using Tukey’s pairwise multiple comparison test and the *p*-values are shown.

**Figure 2 vaccines-10-00270-f002:**
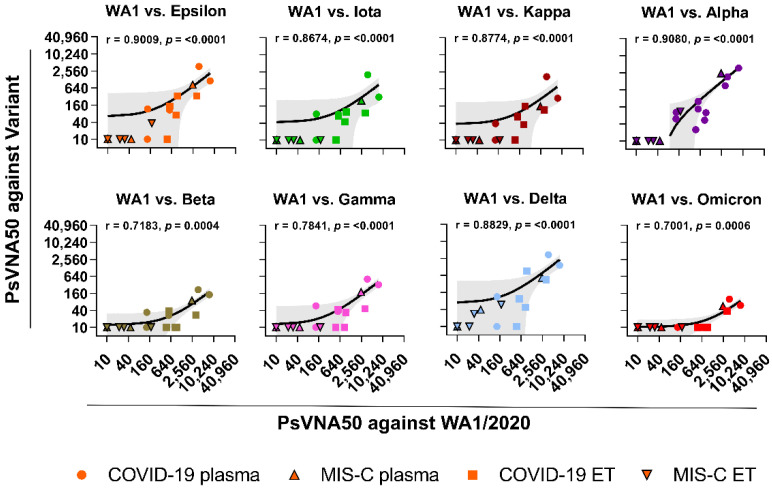
Relationship of neutralizing antibodies against SARS-CoV-2 WA1/2020 and variants. Correlation of SARS-CoV-2 WA1/2020 neutralizing titer versus VOIs; Epsilon, Iota, and Kappa, and VOCs; Alpha, Beta, Gamma, Delta, and Omicron variant neutralizing titer for plasma and endotracheal aspirates of pediatric patients with either COVID-19 (*n* = 5) or MIS-C (*n* = 5). Individual COVID-19 plasma samples are shown as circles, while MIS-C plasma samples are shown as triangles. The endotracheal aspirates samples from pediatric COVID-19 are shown as squares, while MIS-C are shown as inverted triangles. The black line in the scatter plots depicts the linear fit of log2 transformed PsVNA50 values with shaded area showing the 95% confidence interval. Associated correlations show spearman rank correlation coefficients (r) and two-tailed *p* values.

**Table 1 vaccines-10-00270-t001:** Amount of total IgM, IgG, and IgA in undiluted plasma and tracheal samples from pediatric COVID-19 and MIS-C patients.

ANTIBODY ISOTYPES	ACUTE COVID-19 (*n* = 5)		MIS-C (*n* = 5)	
	PLASMA	TRACHEA	PLASMA	TRACHEA
	(mg/mL)	(mg/mL)	(mg/mL)	(mg/mL)
IgM	2.06 ± 0.58	0.051 ± 0.036	1.72 ± 0.34	0.039 ± 0.018
IgG	12.49 ± 3.27	0.231 ± 0.104	13.18 ± 2.72	0.078 ± 0.042
IgA	2.51 ± 1.04	0.047 ± 0.029	2.41 ± 0.69	0.032 ± 0.015

The amount of total IgM, IgG and IgA was quantified using human antibody quantitation kit. Values shown are mean ± standard error of mean.

## Data Availability

All data needed to evaluate the conclusions in the article are present in the manuscript.

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
