# Peer review of "Systemic and Lower Respiratory Tract Immunity to SARS-CoV-2 Omicron and Variants in Pediatric Severe COVID-19 and Mis-C"

_vaccines, 2022, doi:10.3390/vaccines10020270_

Round 1

Reviewer 1 Report

This is one of the best and urgent papers i have read this year. I strongly support its rapid publication in MDPI Vaccines. Some minor and often cosmetic changes the authors could address and incorporate in the revised version are listed below.

1. "Omicron, contains large number of mutations including at least 30 in the SARS-CoV-2 spike protein, raising concerns that Omicron is resistant to neutralizing antibodies generated following SARS-CoV-2 vaccination and infection." I find it is absolutely vital to give some supporting literature sources for this statement: e.g., with regard to escape strategies/ways of omicron from vaccination against other corona variants, etc. The authors can also argue if the number of different corona variants was potentially caused by employment of different vaccines [so that new variants are being created to avoid the antibody response emerged due to the respective vaccines]. Provided the evidence for such a claim exist and are sufficient, this is important to add [in the intro or in the discussion section] also for the future policymakers.

1'. The bibliography itself can be extended to add the most relevant and most closely related studies [both from highly specialized as well as from all-disciplines journals such as Nature and Science]. This, as the experience shows, dramatically enhances the visibility of the future paper and the attention of the community to it. And i would really like this paper getting due attention.

2. Additionally, the mechanisms of this virus-variant escape involving progressively more positive charges on the virus spike-protein-containing surface---from the former variants to the most latest ones---is important to mention, please see and cite Ref. [doi: 10.3934/biophy.2021008]. This enhancement of positive charges on the virus surface due to selectively mutated spike proteins is then involving the electrostatic mechanism of corona-virus-membrane binding to the negatively charged lipid surface of the cell, as thoroughly developed theoretically, see citation ten in the paper above. Mentioning this electrostatic mechanism of binding of DNA as well as lipid-enveloped viruses such as corona to the cell surface seems important for a proper understanding of the first steps of virus attachment to, its fusion with, and penetration of the genetic material into the cell being infected.

3. Regarding the statistical analysis of the data, the authors are encouraged to detail the statements "absolute measurements were log2-transformed before performing the analysis" and "Tukey-adjusted p values were used for denoting significance to reduce Type 1 error due to multiple testing". To do so, i would propose to extend the paragraph 2.4 regarding the sample -collection and statistical-analysis methods and procedures [making it twice as long]. This step is important for confirming the statistical robustness of the results reported [and do so not only for a broad community, but also for real specialists in the field].

4. All acronyms and abbreviations used are to be explained in detail in the text and, at best, also in a special section prior to the bibliography. This will surely facilitate reading of the future manuscript.

5. Red and blue circles in figures 1a and 2 can be hardly distinguishable on black-white printout. Please use additionally to the different colors also different symbols and correlate them [at best, systematically through all the plots] with the variants of corona virus you were studyng. This way the reader will instantly "digest" the information presented in the plots.

6. Regarding the mucosal properties, here the authors are also encouraged to mention some recent particle-tracking studies at variable conditions of mucin-proteins gels, see citation thirteen in the paper above. The penetration of various antigens across the mucosal barriers---in dependence on pH levels of the latter, as observed e.g. in different organs of the body---can be presented in the revised text as a preparatory step of the viral infection, taking place prior to the attachment of virions to the cell membrane. The microscopic aspects of diffusivity and penetration of antigens in general and of corona variants in particular across the mucosal layers as well as through their in vitro reconstituted hydrogels composed of mucin polymers] seems to me important to mention in the revised text.

7. The implications of the main statement of the manuscript "Our data demonstrates that most children and adolescents with severe COVID-19 or MIS-C have weak mucosal neutralizing antibodies in the trachea against circulating SARS-CoV-2 Omicron and other VOCs, and therefore may be susceptible to re-infection with emerging SARS-CoV-2 variants." should be described in more detail. Also the last sentence in the discussion can be extended: elaboration on possible implications of the results obtained for future vaccination strategies---even if the results are preliminary---are extremely important to mention.

Author Response

Reviewer #1 Comments to the Author:

This is one of the best and urgent papers i have read this year. I strongly support its rapid publication in MDPI Vaccines. Some minor and often cosmetic changes the authors could address and incorporate in the revised version are listed below.

Response:

We thank the reviewer for highlighting the importance of our study.

  1. "Omicron, contains large number of mutations including at least 30 in the SARS-CoV-2 spike protein, raising concerns that Omicron is resistant to neutralizing antibodies generated following SARS-CoV-2 vaccination and infection." I find it is absolutely vital to give some supporting literature sources for this statement: e.g., with regard to escape strategies/ways of omicron from vaccination against other corona variants, etc. The authors can also argue if the number of different corona variants was potentially caused by employment of different vaccines [so that new variants are being created to avoid the antibody response emerged due to the respective vaccines]. Provided the evidence for such a claim exist and are sufficient, this is important to add [in the intro or in the discussion section] also for the future policymakers.

Response:

As per reviewer suggestions, we have added references (ref 5-8) and expanded the possible evolution of Omicron in the introduction section.

Lines 80-82: ‘Its suggested that Omicron possibly evolved from ancestral SARS-CoV-2 around mid-2020 or the variant may have evolved in a single immunocompromised individual with long-term SARS-CoV-2 infection[4].’

1'. The bibliography itself can be extended to add the most relevant and most closely related studies [both from highly specialized as well as from all-disciplines journals such as Nature and Science]. This, as the experience shows, dramatically enhances the visibility of the future paper and the attention of the community to it. And i would really like this paper getting due attention.

Response:

We have added multiple references and discussed them further throughout the manuscript.

  1. Additionally, the mechanisms of this virus-variant escape involving progressively more positive charges on the virus spike-protein-containing surface---from the former variants to the most latest ones---is important to mention, please see and cite Ref. [doi: 10.3934/biophy.2021008]. This enhancement of positive charges on the virus surface due to selectively mutated spike proteins is then involving the electrostatic mechanism of corona-virus-membrane binding to the negatively charged lipid surface of the cell, as thoroughly developed theoretically, see citation ten in the paper above. Mentioning this electrostatic mechanism of binding of DNA as well as lipid-enveloped viruses such as corona to the cell surface seems important for a proper understanding of the first steps of virus attachment to, its fusion with, and penetration of the genetic material into the cell being infected.

Response:

We have discussed the possible Omicron escape mechanism.

Lines 231-235: ‘The mechanisms of SARS-CoV-2 variant escape is under investigation that may involve loss of neutralizing antibody epitopes on evolving spike, increase in ACE2 affinity of spike or additional receptor usage or potentially progressively more positive charge on the virus spike-protein resulting in increased viral binding to the negatively charged surface of the cell[6-8, 15, 16].’

  1. Regarding the statistical analysis of the data, the authors are encouraged to detail the statements "absolute measurements were log2-transformed before performing the analysis" and "Tukey-adjusted p values were used for denoting significance to reduce Type 1 error due to multiple testing". To do so, i would propose to extend the paragraph 2.4 regarding the sample -collection and statistical-analysis methods and procedures [making it twice as long]. This step is important for confirming the statistical robustness of the results reported [and do so not only for a broad community, but also for real specialists in the field].

Response:

We have expanded both the samples collection as well as statistical analysis section to clarify further.

Lines 104-107: ’Endotracheal aspirates were suctioned from mechanically ventilated patients using a 6 Fr. catheter attached to the standard trap, kept on ice during transfer to the lab where it was vortexed and then centrifuged at 2500 x g for 10 min. Supernatant was removed and frozen at -80C.’

Lines 161-175: ‘Descriptive statistics were performed to determine the geometric mean titer values and were calculated using GraphPad. All experimental data to compare differences among groups were analyzed using lme4 and emmeans packages in R (RStudio version 1.1.463). The demographic characteristics of these study participants are shown in Table S1. Since age can be biologically plausible confounders, data were analyzed for statistical significance between SARS-CoV-2 strains to control for age as covariate (predictor variables) using a multivariate linear regression model in R. To ensure robustness of the results, absolute neutralization titers were log2-transformed before performing the analysis. For comparisons between the SARS-CoV-2 strains (factor variable), pairwise comparisons were extracted using ‘emmeans’ and Tukey-adjusted p values were used for denoting significance to reduce Type 1 error due to multiple testing in R (RStudio version 1.1.463). The tests were two-sided tests. The differences were considered statistically significant with a 95% confidence interval when the p value was less than 0.05. (* ≤0.05, ** ≤0.01, *** ≤ 0.001, **** ≤0.0001).

Correlation and regression analyses were performed by computing Spearman’s rank correlation coefficient and significance in GraphPad Prism.’

  1. All acronyms and abbreviations used are to be explained in detail in the text and, at best, also in a special section prior to the bibliography. This will surely facilitate reading of the future manuscript.

Response:

The acronyms have been defined at the time of their first appearance and listed prior to the bibliography.

  1. Red and blue circles in figures 1a and 2 can be hardly distinguishable on black-white printout. Please use additionally to the different colors also different symbols and correlate them [at best, systematically through all the plots] with the variants of corona virus you were studyng. This way the reader will instantly "digest" the information presented in the plots.

Response:

The figures have been revised with different shape and color symbols as suggested.

  1. Regarding the mucosal properties, here the authors are also encouraged to mention some recent particle-tracking studies at variable conditions of mucin-proteins gels, see citation thirteen in the paper above. The penetration of various antigens across the mucosal barriers---in dependence on pH levels of the latter, as observed e.g. in different organs of the body---can be presented in the revised text as a preparatory step of the viral infection, taking place prior to the attachment of virions to the cell membrane. The microscopic aspects of diffusivity and penetration of antigens in general and of corona variants in particular across the mucosal layers as well as through their in vitro reconstituted hydrogels composed of mucin polymers] seems to me important to mention in the revised text.

Response:

While this is interesting hypothesis, however, this discussion is out of the purview of this manuscript. Since this study did not investigate any viral infection in different tissues or viral loads in different organs.

  1. The implications of the main statement of the manuscript "Our data demonstrates that most children and adolescents with severe COVID-19 or MIS-C have weak mucosal neutralizing antibodies in the trachea against circulating SARS-CoV-2 Omicron and other VOCs, and therefore may be susceptible to re-infection with emerging SARS-CoV-2 variants." should be described in more detail. Also the last sentence in the discussion can be extended: elaboration on possible implications of the results obtained for future vaccination strategies---even if the results are preliminary---are extremely important to mention.

Response:

We have expanded the discussion.

Lines 225-229: ‘Our data demonstrates that most children and adolescents with severe COVID-19 or MIS-C have weak mucosal neutralizing antibodies in the trachea against circulating SARS-CoV-2 Omicron and other VOCs, and therefore may be susceptible to re-infection with emerging SARS-CoV-2 variants as has been observed in recent months for infection of children with the Omicron variant[14].’

Lines 236-240: ‘The low mucosal immune response in these children can be possibly due to timing of collection of samples at early time-point following hospitalization prior to the generation of strong immune response following infection. In an earlier study, higher neutralizing antibodies against SARS-CoV-2 ancestral strain were observed in bronchoalveolar lavage fluid collected from SARS-CoV-2 adult patients between days 4 and 23 after symptom onset[17].’

Lines 251-254: ‘Therefore, vaccine platforms that can generate effective durable mucosal immunity both in upper and lower respiratory tract would be helpful to provide both protection from disease as well as reduce viral titers in respiratory tract and limit viral transmission to curtail the spread of SARS-CoV-2.’

Reviewer 2 Report

Tang et al investigated the mucosal immunity of pediatric COVID-19 and MIS-C and found that children with severe COVID-19 or MIS-C had weak mucosal neutralizing antibodies in the trachea against circulating SARS-CoV-2 Omicron and other VOCs. Comments are suggested to improve clarity. 

  1. Line 44, please update reference {Ravichandran, 2021 #179}.
  2. Please add references on neutralization of Omicron.
  3. The fold changes in Figure 1B are meaningless. Among most of the neutralization titers against VOCs, only one animal had >20 titer. Was the red triangle from the same patient? Please remove the fold-changes and describe the positive samples.  
  4. The authors found that Neu titers in the endotracheal aspirate was lower than that of plasma in children. Are there any literature indicating that the same is true in adults? Based on the total IgG /A data, it seems that the total antibody amount in endotracheal aspirate was lower than that if plasma (50-over 100 folder lower).

Author Response

Reviewer 2 comments to the Author:

Tang et al investigated the mucosal immunity of pediatric COVID-19 and MIS-C and found that children with severe COVID-19 or MIS-C had weak mucosal neutralizing antibodies in the trachea against circulating SARS-CoV-2 Omicron and other VOCs. Comments are suggested to improve clarity.

    Line 44, please update reference {Ravichandran, 2021 #179}.

Response: Changed reference.

    Please add references on neutralization of Omicron.

Response: Added several references (5-8).

    The fold changes in Figure 1B are meaningless. Among most of the neutralization titers against VOCs, only one animal had >20 titer. Was the red triangle from the same patient? Please remove the fold-changes and describe the positive samples. 

Response: Removed fold-change in fig 1B. The red triangle is same patient sample. We clarified this further.

Lines 212-215: ‘The teen with acute COVID-19 with highest ET neutralization titer (1:2495) against WA1/2020 was obese but otherwise healthy and was intubated 8 days with the ET sample collected on day 4 and showed high neutralizing antibodies against all the variants tested (Supplementary Table 1).’

    The authors found that Neu titers in the endotracheal aspirate was lower than that of plasma in children. Are there any literature indicating that the same is true in adults? Based on the total IgG /A data, it seems that the total antibody amount in endotracheal aspirate was lower than that if plasma (50-over 100 folder lower).

Response:

We have discussed this further and added a recent reference for study in adults.

Lines 236-240: ‘The low mucosal immune response in these children can be possibly due to timing of collection of samples at early time-point following hospitalization prior to the generation of strong immune response following infection. In an earlier study, high neutralizing antibodies against SARS-CoV-2 ancestral strain were observed in bronchoalveolar lavage fluid collected from SARS-CoV-2 adult patients between days 4 and 23 after symptom onset[17].’